# Leukapheresis in Pediatric Acute Leukemia with Hyperleukocytosis: A Single-Center Experience

**DOI:** 10.3390/children9040503

**Published:** 2022-04-02

**Authors:** Sandra Renee Jones, April Rahrig, Amanda J. Saraf

**Affiliations:** 1Department of Internal Medicine, University of Miami and Jackson Memorial Hospital Internal Medicine Residency Program, Miami, FL 33136, USA; sandra.jones@jhsmiami.org; 2Division of Pediatric Hematology & Oncology, Riley Hospital for Children, Indiana University School of Medicine, Indianapolis, IN 46202, USA; alrahrig@iu.edu

**Keywords:** leukostasis, leukapheresis, acute leukemia, hyperleukocytosis, children

## Abstract

Hyperleukocytosis in pediatric acute leukemia is associated with increased morbidity and mortality and at present there is no consensus on the use of leukapheresis (LPH) for its management. Our aim was to review characteristics and outcomes of newly diagnosed leukemia patients with hyperleukocytosis (HL) comparing those who received LPH and those who did not. An IRB approved retrospective case control study reviewed data from a single institution over a 10 year period. At our institution, LPH was used in 8 of 62 (13%) patients with hyperleukocytosis with minimal complications. Mean leukocyte count in patients who received LPH versus those who did not was 498 k cells/mm^3^ and 237 k cells/mm^3^, respectively. Patients who had symptoms of neurologic (63 vs. 17%) or pulmonary leukostasis (75 vs. 17%) were more likely to have undergone leukapheresis. The time from presentation to the initiation of chemotherapy was not different between those who received LPH and those who did not (mean of 35 h vs. 34 h). There was one death in the LPH group, that was the result of neurologic sequelae of hyperleukocytosis and not LPH itself. The use of LPH in patients with hyperleukocytosis is safe, well tolerated and does not alter time to chemotherapy at our institution.

## 1. Introduction

Hyperleukocytosis (HL) is commonly defined as a leukocyte count of 100 k cells/mm^3^ or greater. Morbidity and mortality are increased in cases of HL caused by acute lymphoblastic leukemia (ALL) or acute myeloid leukemia (AML). HL is noted at the time of diagnosis in 5 to 20% of patients with acute leukemia [1]. Patients who present with HL are at an increased risk of developing symptomatic leukostasis [2]. HL is considered an oncologic emergency as these symptoms can present as intracerebral hemorrhage or pulmonary stasis, among others, and can lead to death [2,3,4,5].

In pediatric acute leukemia, HL is more common in ALL than AML [3]. However, children with hyperleukocytic AML are more likely to experience symptomatic leukostasis as well as clinically significant metabolic changes related to tumor lysis [2,5,6,7,8]. Treatment of HL and symptomatic leukostasis involves various measures of leukoreduction. This can include intravenous hydration, initiation of chemotherapy, and leukapheresis [5,8]. The management of HL-associated tumor lysis also includes hydration, as well as medical management hyperuricemia with allopurinol or rasburicase, correction of other electrolyte abnormalities, and in certain cases, renal replacement therapy [4,5].

Leukapheresis is a rapid, effective strategy to reduce peripheral leukocyte counts, however, its use is often contested and controversial. The literature is similarly varied, with published reports ranging from no improvement in early and overall survival to those suggesting improved early survival [4,6,7,9,10,11,12,13]. Other works caution the use of LPH because of high rates of LPH-related complications, while others claim low complication rates and suggest this procedure is safe and well-tolerated [8,14,15,16,17,18]. Though there are no standard of care recommendations for its use, LPH has been utilized as an adjunctive cytoreductive therapy most commonly when patients require reduction of *symptomatic* leukostasis and in an attempt to limit further issues related to metabolic derangements associated with HL [5,14]. In this study, we aim to describe the characteristics of pediatric patients with acute leukemia presenting with hyperleukocytosis, the clinical course of patients who received LPH, and their outcomes compared to those who did not receive LPH at a single institution.

## 2. Materials and Methods

### 2.1. Patients

An IRB-approved, retrospective chart review was completed in all patients with previously untreated acute leukemia between 2009 and 2019 at Riley Hospital for Children at Indiana University who presented with a WBC of greater than 100 k cells/mm^3^. Patients were split into groups based on receiving LPH or not (controls). Data collection included diagnostic and laboratory studies and interventions within the first 72 h of diagnosis, leukapheresis results, and survival outcomes.

### 2.2. Definition of Terms and Complications

Demographic and laboratory data were captured within 72 h of admission to the hospital at the time of diagnosis. Hyperkalemia was defined as serum potassium concentration > 5.5 mmol/L, hyperphosphatemia as >5.2 mg/dL, hypocalcemia as <8.5 mg/dL, hypermagnesemia as >3.0 mg/dL, and hyperuricemia as >7.0 mg/dL, as defined by our institution’s electronic medical record. The relevant maximum or minimum values of potassium, phosphate, calcium, magnesium, and uric acid were recorded and analyzed.

### 2.3. Statistical Analysis

Statistical analysis was performed utilizing Fisher’s exact or *t*-test, where appropriate. Logistic regression was used to generate odds ratios (OR) and 95% confidence intervals (CI) to find predictors of leukapheresis. *p*-values < 0.05 were considered statistically significant. Statistics were performed using Excel 2016 Analysis ToolPak.

## 3. Results

### 3.1. Patient Characteristics

During the study period, 536 children were diagnosed with acute leukemia, of which 62 (12%) had HL. Eight of these 62 patients (13%) received LPH. The clinical presentation of these patients is in Table 1. The mean age was 13 years (range 7–22 years) in the leukapheresis group and 8 years (range 0–24 years) in the control group. The mean leukocyte count in the leukapheresis group was 498 k cells/mm^3^ (range 324–751 k cells/mm^3^) and 237 k cells/mm^3^ (range 102–659 k cells/mm^3^) in the control group (*p* = 0.01). The mean time from presentation to starting chemotherapy was 35 h (range 6–61 h) in the LPH group and 34 h (2–62 h) in the control group (*p* = 0.932). One patient received chemotherapy prior to leukapheresis. The mean duration of ICU (intensive care unit) stay was not significantly different between the two groups (13 days for LPH and 2 days for the control group, *p* = 0.1). The mean duration of initial hospital stay was also not significantly different between the two groups (35 days in the LPH group and 16 days in the control group (*p* = 0.3).

### 3.2. Signs and Symptoms of Leukostasis

Five (63%) patients in the LPH group and 9 (17%) in the control group presented with at least one sign or symptom of neurologic leukostasis, such as vision changes, altered mental status, or intracranial hemorrhage (*p* = 0.01). Six (75%) of the LPH group and 9 (17%) of the controls had pulmonary leukostasis, such as dyspnea, hypoxia, alveolar hemorrhage, or respiratory failure (*p* = 0.01). All patients in the LPH group and 15 (28%) in the control group presented with either neurologic or pulmonary leukostasis (*p* = 0.0001). Three (38%) of the LPH group and 2 (4%) of the control group had both neurologic and pulmonary leukostasis (*p* = 0.02). Clinical leukostasis data is presented in Table 1.

### 3.3. Mortality Outcomes

One patient (13%) in the LPH group and no patients in the control group died in the first 30 days of presentation. This patient in the LPH group (patient one in Table 2) was transferred from an outside hospital and went into cardiac arrest shortly after arrival. After the return of spontaneous circulation was achieved, a post-arrest head computerized tomography showed multifocal intracranial hemorrhage and midline shift. Early the next morning, apheresis was completed and post-procedure scans showed stable midline shift, new foci of hemorrhage, and bilateral uncal herniation. Later that morning this patient was pronounced brain dead and succumbed to cardiac death after ventilatory support was discontinued.

### 3.4. Predictors of Leukapheresis

Patient characteristics as predictors of the odds of undergoing leukapheresis are enlisted in Table 3. In logistic regression analysis, increasing age (OR 1.04, *p* = 0.01), Asian decent (OR 26.0, *p* = 0.02), elevated leukocyte count (OR 1.003, *p* = 1 × 10^−5^), neurologic leukostasis (OR 45.7, *p* = 0.01), pulmonary leukostasis (OR 54.1, *p* = 0.01), one or more symptom of neurologic or pulmonary leukostasis (OR 43.3, *p* = 0.01), and both neurologic and pulmonary leukostasis (OR 110.6, *p* = 0.01) increased the odds of receiving leukapheresis. No significant association with leukapheresis was found in sex, immunophenotype, cytogenetics, CNS (central nervous system) disease, or mortality.

### 3.5. Complications of Leukapheresis

Four patients (50%) who received LPH had a complication related to leukapheresis. One complication was of mild intra-procedural hypotension and three patients experienced hypocalcemia (Table 2). Those patients with hypocalcemia were managed only with calcium repletion. No active intervention was required for intra-procedural hypotension, which resolved after LPH completion.

## 4. Discussion

Hyperleukocytosis is an oncologic emergency in acute leukemia with significant and potentially fatal side effects. As a result of HL, neurologic and pulmonary leukostasis have been shown to increase the risk of mortality [2,3]. Diagnosing and treating symptomatic leukostasis is imperative to mitigating these risks. Diagnosis of leukostasis, while clinical, accounts for the degree of WBC elevation. Our cohort receiving leukapheresis had a significantly higher leukocyte count than those who did not receive this intervention, an observation consistent with the literature [2]. Other studies have recommended cytoreduction with LPH in patients with ALL with leukocyte counts over 400 k cells/mm^3^, and our institution did not perform LPH in on patients with leukocyte counts less than 300 k cells/mm^3^ [2].

Our institution saw no difference in time to chemotherapy initiation in patients who received LPH versus those who did not. It was shown previously that leukapheresis does not delay the time to chemotherapy [2,6]. Most patients who received LPH did so before the initiation of chemotherapy, but this did not result in differences between groups.

There was one patient who succumbed to early death in the LPH group. While their clinical picture further deteriorated after LPH, it was determined that this was due to CNS hemorrhage secondary to hyperleukocytosis and not LPH itself.

Increasing age, male sex, T-cell ALL phenotype, and M4/M5 AML subtypes are predictors for LPH [3,4,8,14]. In our study, we did not note these associations. However, age and Asian ethnicity were associated with increased odds of LPH. Ethnicity data has not yet been established as a predictor in the literature. We hypothesize that this ethnicity data reflects either our small cohort size with only three patients who were Asian or social implications of healthcare and/or healthcare disparities rather than innate biologic risk factors.

Clinical symptoms of leukostasis increase the odds of receiving LPH [4,15]. In our study, we saw that all metrics of leukostasis (neurologic, pulmonary, either neurologic or pulmonary, and neurologic and pulmonary) between groups or through logistic regression were more common in the group receiving leukapheresis compared to the control group. All patients who had leukapheresis presented with a sign or symptom of leukostasis, a feature not shared by the controls [4]. This outcome likely reflects that physicians at our institution initiate LPH based on the clinical scenario rather than directly in response to hyperleukocytosis. Our research did not demonstrate statistical differences across ICU length of stay and rates of mechanical ventilation in patients who received LPH, in contrast to previously reported studies [4]. In alignment with other investigations, those receiving LPH had higher rates of mechanical ventilation, but this finding was not statistically significant [4]. Performing the leukapheresis procedure has inherent risks. Previously reported procedural risks include hemodynamic changes, electrolyte derangements, bleeding related to the use of anticoagulation, and risks associated with central lines [5,6]. However, our institution and many others have reported minimal to no significant complications with this procedure [2,6,19].

Overall, patients presenting with hyperleukocytosis are at risk of developing significant CNS and pulmonary side effects. These manifestations can persist despite supportive interventions. Leukapheresis is a safe tool that can augment symptom management and potentially prevent significant and life-threatening complications while not delaying the time to initiation of chemotherapy. At our institution, the use of leukapheresis was without life-threatening complications and did not delay the initiation of chemotherapy. Conclusively, these findings suggest that leukapheresis is safe, well-tolerated, and utilized in patients with symptomatic hyperleukocytosis. Our results are limited by the small cohort size and its retrospective design. Future works should consider looking prospectively at the role of leukapheresis among all the leukocytoreductive techniques in hyperleukocytic acute leukemia in children.

## Figures and Tables

**Table 1 children-09-00503-t001:** Characteristics of patients who did or did not receive leukapheresis within 72 h of presentation.

Characteristics	LPh ^a,b^ (*n* = 8)	No LPh ^a,b^ (*n* = 54)	*p*
Age * (years)	13 ± 5 (7, 22)	8 ± 6 (0, 24)	0.020
Sex			0.573
Male	6 (75%)	35 (65%)	
Female	2 (25%)	19 (35%)	
Race *			0.014
White	3 (38%)	41 (76%)	
Latinx	1 (13%)	7 (13%)	
Asian	2 (25%)	1 (2%)	
Black	2 (25%)	5 (9%)	
Immunophenotype			0.111
ALL	5 (63%)	47 (90%)	
AML	3 (38%)	7 (13%)	
CNS disease	2 (33%)	16 (30%)	0.767
Cytogenetics			1.000
Abnormal but not significant	6 (75%)	39 (72%)	
Normal	2 (25%)	15 (28%)	
Complete blood count			
WBC ** (k cells/mm^3^)	498.2 ± 160.8 (324.0, 751.0)	236.6 ± 123.5 (101.8, 659.0)	0.001
Hemoglobin (g/dL)	8.1 ± 2.9 (8.3, 12.9)	7.6 ± 2.6 (6.1, 11.7)	0.685
Platelet (k/dL)	54 ± 32 (14, 99)	50 ± 49 (6, 317)	0.752
Electrolyte abnormalities			1.000
Tumor lysis syndrome	6 (75%)	40 (74%)	
No tumor lysis syndrome	2 (25%)	14 (26%)	
Findings, signs, and symptoms			
Neurologic leukostasis *	5 (63%)	9 (17%)	0.011
Pulmonary leukostasis **	6 (75%)	9 (17%)	0.002
N or P leukostasis ***	8 (100%)	15 (28%)	0.001
N and P leukostasis *	3 (38%)	2 (4%)	0.024
Treatment			
Rasburicase	5 (63%)	15 (28%)	0.098
Time to CT (hours)	35 ± 17 (6, 61)	34 ± 14 (2, 62)	0.932
Transfusions in first two weeks of stay			
pRBCs (units)	6 ± 3 (3, 10)	3 ± 3 (0, 13)	0.115
Platelets (units)	9 ± 13 (0, 39)	4 ± 3 (0, 13)	0.303
LOS (days) of initial hospitalization			
Overall	35 ± 40 (3, 132)	16 ± 13 (3, 51)	0.253
Initial stay in PICU	13 ± 23 (2, 72)	2 ± 3 (0, 16)	0.223
Mortality			
30-day Mortality	1 (13%)	0	0.129

ALL = acute lymphoblastic leukemia; AML = acute myeloid leukemia; AMS = altered mental status; CT = chemotherapy; HA = headache; LOS = length of stay; N = neurologic; P = pulmonary; PICU = pediatric intensive care unit; pRBCs = packed red blood cells; WBC = white blood cell. ^a^ mean ± standard deviation (minimum, maximum), ^b^ number (% of subgroup). * = *p* < 0.05, ** = *p* < 0.01, *** = *p* ≤ 0.001.

**Table 2 children-09-00503-t002:** The initial presentation and outcomes of patients who had leukapheresis.

Pt	Age/ Sex	Diagnosis	Initial WBC (k Cells/mm^3^)	Leukostasis Signs and Symptoms	Time to LPH [#]	Complications of LPH	Time to CT	Status/ Days to Death
1	19/M	B-ALL	680	ICH, hypoxia, AMS, HA	34 h [1]	mild hypotension	46 h	Deceased/3
2	9/M	AML	394	respiratory distress, tachypnea, hypoxia	6 h [1]	hypocalcemia	31 h	Deceased/192
3	7/M	T-ALL	751	SOB, HA	12 h [2]	hypocalcemia	44 h	Alive
4	8/M	B-ALL	521	hearing loss, AMS	23 h [1]	none	36 h	Alive
5	14/F	T-ALL	580	LOC, cerebellar ataxia, AMS, ICH	10 h [3]	none	6 h	Alive
6	22/F	AML	339	HA, blurry vision	14 h [2]	none	61 h	Alive
7	14/M	AML	398	HA, blurry vision	6 h [2]	none	41 h	Alive
8	13/M	B-ALL	324	hypoxia, HA, AMS	8 h [1]	hypocalcemia	12 h	Alive

ALL = acute lymphoblastic leukemia; AML = acute myeloid leukemia; AMS = altered mental status; CT = chemotherapy; F = female; HA = headache; ICH = intracerebral hemorrhage; LOC = loss of consciousness; LPH = leukapheresis; M = male; SOB = shortness of breath; WBC = white blood cell.

**Table 3 children-09-00503-t003:** Patient characteristics as predictors of the odds of undergoing leukapheresis.

Characteristics	Odds Ratio	95% CI (Lower, Higher)	*p*-Value
Age (continuous) *	1.040	1.008, 1.074	0.014
Male vs. Female	1.629	0.299, 8.870	0.573
Race (compared to White)			
Latinx	1.857	0.168, 20.512	0.614
Asian *	26.000	1.796, 376.321	0.017
Black	5.200	0.692, 39.081	0.109
WBC (continuous) ***	1.003	1.002, 1.004	0.001
Immunophenotype			
AML vs. ALL	4.029	0.784, 20.703	0.095
T-cell vs. B-cell	1.074	0.163, 7.063	0.941
CNS Disease	1.156	0.192, 6.966	0.874
Cytogenetics and Molecular Studies (compared to normal cytogenetics)			
FLT3	7.500	0.645, 87.197	0.107
KMT2A-R	0.365	0.016, 8.507	0.530
ETV6-RUNX1	0.689	0.028, 17.116	0.820
BCR-ABL	0.886	0.034, 22.853	0.942
ABL mutation	1.500	0.111, 20.300	0.760
CDKN2A	1.875	0.134, 26.321	0.641
Abnormal but not significant	1.154	0.142, 9.385	0.894
Leukostasis (compared to no P or N)			
Neurologic *	45.737	2.322, 900.898	0.012
Pulmonary **	54.053	2.794, 1045.619	0.008
N or P *	43.323	2.355, 796.926	0.011
N and P **	110.600	4.387, 2788.059	0.004
Mortality	1.917	0.327, 11.227	0.471

ALL = acute lymphoblastic leukemia; AML = acute myeloid leukemia; CI = confidence interval; N = neurologic; P = pulmonary; WBC = white blood cell. * = *p* < 0.05, ** = *p* < 0.01, *** = *p* ≤ 0.001.

## Data Availability

Data is available upon reasonable request.

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
