# Peer review of "Leukapheresis in Pediatric Acute Leukemia with Hyperleukocytosis: A Single-Center Experience"

_children, 2022, doi:10.3390/children9040503_

Round 1

Reviewer 1 Report

The hyperleukocytosis is combined with high risk of developing significant CNS and pulmonary side effects. Searching for new methods of supportive interventions is necessary. Leukapheresis is a safe, well-tolerated method,  that can augment symptom management, prevent all serious complications, and does not delay the time to induction therapy.

The above thesis is justified in the paper - in the designed methodology, presented results, and conducted discussion. The authors are critical of the obtained results e.i. small cohort size and the retrospective character of work.

In my opinion, correctly prepared paper.

Minor revisions:
1. In the section: introduction : I proposed to expand the topic concerning the methods of dealing with hyperleukocytosis - medications, methods of preventing the TLS, reducing leukocytosis.
2. In the section: discussion: I encourage authors to discuss the diseases that occur with hyperleukocytosis in the context of the discussion with their own results.

Author Response

Thank you kindly for your review of our paper and your thoughtful comments. 

  1. In the section: introduction : I proposed to expand the topic concerning the methods of dealing with hyperleukocytosis - medications, methods of preventing the TLS, reducing leukocytosis.

We appreciate this suggestion and have expanded this topic within the introduction.

  1. In the section: discussion: I encourage authors to discuss the diseases that occur with hyperleukocytosis in the context of the discussion with their own results.

Thank you for this suggestion.  We have attempted to augment this discussion throughout the manuscript.

Reviewer 2 Report

This is a well written article with sound methodology, results are well displayed. The tables add a lot to the article and are easy to follow. This article is a retrospective review at a single institution highlighting leukapheresis usage in eight pediatric patients with acute leukemia. The article is descriptive and uses a control group for comparisons to the LPH group. The authors conclude that there is no difference in outcomes between groups and that LPH is safe for management of hyperleukocytosis. This is a controversial topic in the literature as there is mixed evidence of efficacy of LPH and this article adds evidence of its safety in pediatric patients.

General comments:

-It would be helpful to describe the risks of LPH.

-It may be helpful to add why some would recommend leukapheresis instead, or in addition to, chemotherapy for leukoreduction in patients with hyperleukocytosis.

-It would be helpful to state that HL is an oncologic emergency somewhere in the introduction.

Specific comments:

31- Tumor lysis not a symptom of leukostasis, but is a symptom of HL

39- Would change sentence structure for sentence “Studies suggest…”

Table 2: Cytogenetics category: Should be cyto/molecular if including FLT3. Also, MLL-R should be KMT2A-R to reflect the new nomenclature. T(12/21) could instead be ETV6-RUNX1 to coordinate with the remaining nomenclature.

151- is there an LPH pathway at your institution? If so, maybe helpful to touch upon decision making surrounding LPH. Or if there are any published recommendations it may be helpful to touch upon them here.

Author Response

We thank you for your careful editing of our manuscript.  We appreciate your recommendations.

General comments:

-It would be helpful to describe the risks of LPH.

 Thank you for pointing out this omission – we have added this to our discussion.

-It may be helpful to add why some would recommend leukapheresis instead, or in addition to, chemotherapy for leukoreduction in patients with hyperleukocytosis.

 We appreciate this suggestion – we have added a statement that while there is no standard of care to direct usage, it is most commonly utilized as a part of the cytoreductive plan in patients that are symptomatic and felt to be at high risk for complications related to tumor lysis/metabolic concerns.

-It would be helpful to state that HL is an oncologic emergency somewhere in the introduction.

 Thank you – we agree that this should be clearly stated earlier and has been added to the introduction.

Specific comments:

31- Tumor lysis not a symptom of leukostasis, but is a symptom of HL

Thank you for noticing that.  This has been corrected within the manuscript.

39- Would change sentence structure for sentence “Studies suggest…”

 We appreciate that feedback.  This sentence has been restructured for readability.

Table 2: Cytogenetics category: Should be cyto/molecular if including FLT3. Also, MLL-R should be KMT2A-R to reflect the new nomenclature. T(12/21) could instead be ETV6-RUNX1 to coordinate with the remaining nomenclature.

 This section has been edited to reflect these changes.

151- is there an LPH pathway at your institution? If so, maybe helpful to touch upon decision making surrounding LPH. Or if there are any published recommendations it may be helpful to touch upon them here.

This is an excellent question.  We do not have an LPh pathway at our institution, but we are in the process of formalizing one at present.  There are no recommendations in the literature that are accepted as standard of care.